# Anti-Biofilm Activities of Chinese Poplar Propolis Essential Oil against *Streptococcus mutans*

**DOI:** 10.3390/nu14163290

**Published:** 2022-08-11

**Authors:** Jie Yuan, Wenqin Yuan, Yuyang Guo, Qian Wu, Fei Wang, Hongzhuan Xuan

**Affiliations:** School of Life Sciences, Liaocheng University, Liaocheng 252059, China

**Keywords:** propolis essential oil, *Streptococcus mutans*, dental caries, biofilm, glucosyltransferase, extracellular polysaccharides, cytotoxicity

## Abstract

*Streptococcus mutans* (*S. mutans*) is a common cariogenic bacterium that secretes glucosyltransferases (GTFs) to synthesize extracellular polysaccharides (EPSs) and plays an important role in plaque formation. Propolis essential oil (PEO) is one of the main components of propolis, and its antibacterial activity has been proven. However, little is known about the potential effects of PEO against *S. mutans*. We found that PEO has antibacterial effects against *S*. *mutans* by decreasing bacterial viability within the biofilm, as demonstrated by the XTT assay, live/dead staining assay, LDH activity assay, and leakage of calcium ions. Furthermore, PEO also suppresses the total of biofilm biomasses and damages the biofilm structure. The underlying mechanisms involved may be related to inhibiting bacterial adhesion and GTFs activity, resulting in decreased production of EPSs. In addition, a CCK8 assay suggests that PEO has no cytotoxicity on normal oral epithelial cells. Overall, PEO has great potential for preventing and treating oral bacterial infections caused by *S. mutans*.

## 1. Introduction

Dental caries is one of the most common oral infectious diseases and is expensive to manage [1]. The occurrence of dental caries is closely related to microorganisms, diet such as the intake of sugary food and beverages, and the host’s oral hygiene habits. The formation of dental biofilm dominated by microorganisms is the virulence factor initiating dental caries. If left untreated, dental caries leads to pulpitis, periapical inflammation, and even loss of teeth [2].

The genus *Streptococcus* accounts for a high proportion of all supragingival microorganisms present in the oral biofilm, *Streptococcus mutans* (*S. mutans*), being one of the main pathogens that plays a crucial role in the etiology of human dental caries and peri-implant infections [3]. *S. mutans*, which survives in the plaque biofilm, can rapidly ferment a variety of carbohydrates to produce a large amount of acid, reducing the local pH and resulting in the fall-off of local hard tissue and the beginning of pathological caries. Furthermore, *S. mutans* can secrete glucosyltransferases (GTFs) and fructosyltransferase (FTF) [4]. These enzymes synthesize extracellular polysaccharides (EPSs), mainly glucan and fructan, using dietary sucrose as a substrate, damaging the tooth enamel and causing the formation and accumulation of plaque or dental biofilm [5,6].

Mechanical methods using toothbrushes and dental floss are traditionally applied to the removal of dental plaque. In addition, antibacterial agents such as chlorhexidine are often chosen for the control and prevention of dental caries [7]. However, the frequent use of antibiotics may also lead to increased bacterial resistance [8]. Therefore, it is necessary to find more effective products to prevent and treat plaque; an increasing number of studies have shown that natural products play a more important role in caries treatment [9,10,11].

Propolis is a complex material that honey bees collect from resinous and balsamic material, which has a variety of biological activities, such as antibacterial [12], anti-tumor [13], antioxidant [14], anti-inflammatory [15,16], immunomodulatory [17], etc. It has been used since ancient civilizations to alleviate many ailments, including pathogenic infections. As a natural antimicrobial compounds, propolis is widely used in toothpaste, oral sprays, chewing gums and other daily products to protect against oral cavities. Propolis essential oil (PEO) is one of the main components of propolis, accounting for about 10% [18]. Its antibacterial activity to Gram-negative and Gram-positive bacteria has been proven [19]. However, there is no detailed study on the effects of PEO on dental caries. We aimed to investigate PEO’s antibacterial and anti-biofilm activities against the pioneer Gram-positive colony of oral cavities, *S. mutans*, and the major mechanisms of PEO in suppressing biofilm.

## 2. Materials and Methods

### 2.1. Propolis Sample and Bacterial Strain

Propolis was harvested from Shandong province in Eastern China in 2020, with poplar (*Populus* sp.) being the main plant of origin. *S. mutans* (ATCC 25175) was acquired from Guangdong Microbiology Culture Center and inoculated into brain–heart infusion broth (BHIB; Hopebio, Qingdao, China). After incubation, strains were maintained in brain–heart infusion agar (BHIA) at 4 °C and 20% glycerol at −80 °C for longer preservation. Human oral epithelial cells (HOECs) were purchased from Cell Bank of Typical Culture Preservation Committee, Chinese Academy of Sciences, Shanghai (Shanghai, China). Cells were cultured in DMEM supplemented with 10% (*v*/*v*) fetal calf serum (FBS) and 100 U/mL of penicillin as well as 100 μg/mL streptomycin. The cell culture conditions were in 5% CO_2_ at 37 °C.

### 2.2. Preparation of PEO

Propolis samples (30 g) were freeze-dried and crushed. According to relevant report [10], the fragments obtained were hydro-distilled for 6 h using a Clevenger-type apparatus. The resulting essential oils were sealed and preserved at −20 °C until use. The stock solution was prepared by dissolving 5 μL of PEO in 1 mL of BHIB medium containing 1% Dimethyl sulfoxide (DMSO). The working solution of PEO was prepared by diluting it in BHIB to maintain the final DMSO concentration of 1%.

### 2.3. Gas Chromatography-Mass Spectrometry (GC–MS) Analysis of PEO

PEO components were analyzed on a GC-2010 Plus Gas chromatography-mass (Shimadzu Co., Tokyo, Japan) equipped with an HP-5MS UI column (30 m × 0.25 mm, 0.25 μm) as described previously [20,21]. The carrier gas was helium, and the flow rate was 1.0 mL/min. The heating procedure included an initial oven temperature of 40 °C for 5 min, then 10 °C/min to 80 °C, 3 °C/min to 210 °C, and 30 °C/min to 300 °C. The injection temperature was 260 °C, and the split ratio was 50/1.

### 2.4. Antibacterial Activity Assessment

#### 2.4.1. Determination of Inhibition Zone Diameter (DIZ)

The method of DIZ was performed according to a published report [22]. In brief, 100 μL of logarithmic growth phase bacterial suspension (10^7^ CFU/mL) was spread onto the BHIA medium. Sterile filter paper disks (6 mm) were treated with PEO to achieve a final concentration of 10 μL/disc; then, the filter paper was placed onto the surface of the BHIA medium and incubated at 37 °C for 24 h.

#### 2.4.2. Determination of the Minimal Inhibitory Concentration (MIC)

The resazurin microdilution method determined the MIC of PEO against *S. mutans* [23]. A total of 100 μL of PEO was mixed with 100 μL of the bacterial suspension (10^7^ CFU/mL) in a 96-well plate, obtaining the final PEO concentrations ranging from 0.156 to 2.5 μL/mL. The plates were incubated at 37 °C for 24 h. Finally, 20 μL of resazurin sodium (1 mg/mL) was added to each well and incubated at 37 °C for 3 h in the dark. BHIB containing only 1% DMSO (control) was also observed. The MIC was the lowest concentration of PEO preventing the solution from turning from blue to pink.

#### 2.4.3. Determination of the Minimum Bactericidal Concentration (MBC)

The MBC of PEO against *S. mutans* was determined by the BHIA dilution method. We transferred the bacterial suspension treated with PEO, where there was no bacterial growth in the above wells, onto BHIA plates and subcultured it for another 24 h at 37 °C. MBC is defined as the minimum concentration for which no bacterial colonies were seen on the agar plate after 24 h of incubation.

#### 2.4.4. Determination of Growth Curve

The effects of PEO on the growth of *S. mutans* were determined as described previously [24]. A bacterial suspension of 500 μL (10^7^ CFU/mL) was inoculated into a tube containing a 5 mL BHIB medium and adjusted to an optical density of 600 nm (OD_600_) at 0.2. Afterwards, the bacterial suspensions w exposed to different concentrations of PEO from 1/2 MIC to 2 MIC and incubated at 37 °C for 24 h. The absorbance value was determined every 2 h at 600 nm using a microplate reader (ELX808, BioTek, Winooski, VT, USA).

### 2.5. Biofilm Assessment

#### 2.5.1. Determination of Total Biofilm Biomasses

Crystal violet (CV) is an alkaline dye that simultaneously stains both living and dead cells; thus, it is often used to measure the total amount of cells in biofilm [10]. A bacterial suspension of 100 μL and 100 μL of BHIB medium supplemented with 1% sucrose were added to a 96-well plate and cultured at 37 °C for 24 h to form biofilm. After the incubation, the supernatant was removed, and the biofilms attached to the plate were gently rinsed three times with sterile phosphate-buffered saline (PBS, pH 6.8). Then, 200 μL of different PEO concentrations ranging from 1/16 MIC to 2 MIC were added to each well and incubated for 24 h at 37 °C. Afterwards, the supernatant was removed, carefully rinsed three times with PBS, and added to 200 μL of methanol to fix the biofilm for 15 min; then, 200 μL of CV solution (1%) was added and maintained for 5 min. The CV solution was removed, and each well was gently washed with distilled water multiple times. Then, a mixed organic reagent of ethanol: acetone = 3:7 was added. The absorbance value was read at 492 nm using a microplate reader (ELX808, BioTek, USA).

#### 2.5.2. Determination of the Cell Activity within Biofilm

The 2,3-bis (2-methoxy-4-nitro-5-sulfophenyl)-5-[(phenylamino) carbonyl]-2H- tetrazolium hydroxide (XTT) can be reduced to water-soluble brown formazan by mitochondrial dehydrogenase of living cells, and the formazan amount is positively correlated with the living cells [25]. Therefore, an XTT reduction assay was used to evaluate the effect of PEO on bacterial activity within biofilm. A bacterial suspension of 100 μL and 100 μL of BHIB supplemented with 1% sucrose were added to a 96-well plate and cultured at 37 °C for 24 h to form biofilm. The supernatant was discarded and gently rinsed three times with a PBS buffer (pH 6.8). Then, 100 μL of BHIB medium and 100 μL of different concentrations of PEO concentrations from 1/16 MIC to 4 MIC were added and cultured at 37 °C for 24 h. *S. mutans* suspension was the negative control group. An XTT reaction solution was mixed with XTT (1 mg/mL dissolved in PBS) and menadione solutions (2 μmol/mL dissolved in acetone) in a volume ratio of 50:1. After incubation, the supernatant was discarded and gently rinsed three times with PBS (pH 6.8). Then, 100 μL of BHIB medium and 100 μL of XTT reaction solution were added to react for 2 h in the dark at 37 °C, and the absorbance values of the supernatants were measured at 450 nm using a microplate reader (ELX808, BioTek, USA).

#### 2.5.3. Live/Dead Bacteria Staining

The Live/Dead^®^BacLight^TM^ Bacterial Viability kit (L13152, Invitrogen, Waltham, MA, USA) was also used to determine the bacterial viability within the biofilm. The kit includes two dyes: SYTO 9, staining all bacterial cells with green fluorescence, and PI, staining cells with impaired membranes with red fluorescence.

A total of 1 mL of bacterial suspension was added to the laser confocal dish and incubated for 24 h at 37 °C Afterward, the supernatant was removed, and carefully rinsed once with PBS (pH 7.4). Different PEO concentrations from 1/2 MIC to 4 MIC in a BHI medium were added to the dishes and incubated for 18 h at 37 °C. After that, we discarded the supernatant and gently washed it twice with sterile water, and then it was stained with Live/Dead^®^BacLight^TM^ Bacterial Viability kit for 15 min in the dark at room temperature. The stained cells were observed under a laser confocal microscope (Olympus FV1200, Kyoto, Japan).

#### 2.5.4. Scanning Electron Microscopy (SEM) Observation of Biofilm

SEM was used to observe the effects of PEO on bacterial biofilm structure, and the detailed method can be found in our previously published paper [10].

### 2.6. Cell Damage Assay

#### 2.6.1. Determination of Lactic Dehydrogenase (LDH) Activity

LDH activity was used to detect bacterial cell damage within the biofilm as described previously [26]. In brief, 100 µL of bacterial suspension and 100 µL of BHIB supplemented with 1% sucrose were added into a 96-well plate and incubated for 24 h at 37 °C to form bacterial biofilms. After the incubation, the supernatant was removed and gently rinsed three times with sterile PBS (pH 6.8). Then, 200 µL of different PEO concentrations of from 1/16 MIC to 4 MIC in a BHIB medium were added and incubated for 24 h at 37 °C; the supernatant was collected to detect LDH activity using a LDH detection kit at 450 nm. Bacterial suspension and BHI medium were used as negative controls.

#### 2.6.2. Leakage of Calcium Ion

Ca^2+^ can also be used to indicate bacterial cells damage within the biofilm [27]. When the integrity of the bacterial cell membrane is destroyed, the content of Ca^2+^ in the supernatant will increase. A total of 100 μL of bacterial suspension and BHIB supplemented with 1% sucrose were added to a 96 well plate and cultured at 37 °C for 24 h. After the biofilm had formed, the supernatant was discarded and gently rinsed three times with PBS (pH 6.8). Then, 100 μL of BHIB medium and 100 μL of different PEO concentrations from 1/16 MIC to 4 MIC were added and cultured at 37 °C for 24 h. The supernatant was collected for Ca^2+^ detection with a Ca^2+^ detection kit (Nanjing Jiancheng Bioengineering Institute) according to the manufacturer’s instructions.

### 2.7. Determination of Bacterial Adhesion

The effect of PEO on bacterial adhesion was measured by comparing the changes in bacterial adhesion with or without PEO treatment, as described previously [28]. A total of 100 μL of bacterial suspension and 100 μL of BHIB supplemented with 1% sucrose were added to a 96-well plate and cultured for 24 h at 37 °C to form biofilms. Then, the above supernatant was discarded and gently rinsed three times with PBS (pH 6.8). Afterward, 200 μL of BHIB medium with or without PEO (from 1/4 MIC to 4 MIC) was added to each well and cultured for another 24 h. The bacterial suspension was used as the negative control group. The supernatant was discarded and carefully rinsed the remaining three times with PBS (pH 6.8). A total of 100 μL BHIB medium was added, an ultrasound was performed before the optical density was tested at 600 nm.

### 2.8. Extracellular Polysaccharides (EPSs) Production Assay

The method for evaluating the effects of PEO on the production of EPSs was carried out according to the published paper [29]. A total of 500 μL of the bacterial suspension and 5 mL BHIB medium, with or without different concentrations of PEO from 1/16 MIC to 4 MIC, were mixed and cultured for 24 h at 37 °C. Then the mixture was centrifuged at 12,000× *g* for 30 min at 4 °C to discard the supernatant, and the sediment was resuspended with sterile water and centrifuged at 12,000× *g* for 30 min at 4 °C to collect the supernatant containing water-soluble polysaccharides.

The sediment was resuspended with NaOH (0.1 mol/L) and centrifuged at 12,000× *g* for 30 min at 4 °C to collect the supernatant and this was repeated three times. These supernatants were combined, and three times the volume of 95% ethanol was added overnight at 4 °C. The mixture was centrifuged at 12,000× *g* for 30 min at 4 °C, and the precipitate was water-insoluble polysaccharides. The precipitate was dissolved with NaOH (0.1 mol/L) to determine its content.

The content of extracellular polysaccharides, including water-soluble polysaccharides and water-insoluble polysaccharides, was measured using the phenol-sulfuric acid method at 492 nm.

### 2.9. The Extraction and Determination of GTFs

The method for producing the crude extract of GTFs was used as described by Liu et al. (2017) [9]. A total of 20 mL of bacterial suspension in the logarithmic growth phase was added to a 200 mL BHIB medium and cultured for 24 h at 37 °C. After incubation, the bacterial suspension was centrifuged at 12,000× *g* for 30 min at 4 °C to collect the supernatant. Then, the supernatant was precipitated using two-thirds of the ammonium sulfate volume (60%). The precipitate was centrifuged at 12,000× *g* for 30 min at 4 °C to remove the supernatant, then dissolved with PBS (pH 6.8) and dialyzed for 48 h with a 50 KD dialysis bag to obtain crude GTFs. A total of 1 mL 0.1 M sucrose was added as the substrate, and 200 μL of crude enzyme, with or without different concentrations of PEO, in a 12-well plate; the mixtures were cultured for 18 h at 37 °C. The change in the water-insoluble polysaccharide content produced was used to indicate PEO’s effect on GTFs. The content of water-insoluble polysaccharides was determined by the method described in Section 2.8.

### 2.10. Determination of PEO Cytotoxicity to HOEC Cells

Cell viability was determined using the CCK8 kit (Genview Scientific Inc, Houston, TX, USA) as described previously [13]. Human oral epithelial cells (HOECs) (1 × 10^5^) were seeded in a 96-well plates. When the cells reached 80% confluence, they were treated with PEO (0.005, 0.05, 0.5, and 5 μL/mL respectively). At 24 h, cell viability was measured according to the manufacturer’s instructions. The optical density was read at 450 nm.

### 2.11. Statistical Analysis

All experiments were repeated at least three times independently. Data are presented as the mean ± SD. SPSS 18.0 software was paired with Student’s *t*-test and ANOVA. The *p*-value < 0.05 was considered statistically significant.

## 3. Results

### 3.1. The Main Constituents of PEO

GC-MS analysis was used to identify the major constituents of PEO, and 79 constituents were identified. Table 1 indicated 16 major constituents with a relative amount of more than 1.0%, including β-himachalene (13.94%), α-curcumene (11.28%), α-bergamotene (4.5%), sesquicineole (4.35%), β-Bisabolene (3%).

### 3.2. The Antibacterial Activities of PEO against S. mutans

DIZ, MIC, and MBC were used to evaluate the antibacterial activities of PEO against *S. mutans* and results indicated that PEO significantly inhibited *S. mutans* (Table 2). The DIZ value of PEO (24.5 mm) was higher than gentamycin (22.5 mm), ampicillin (11.0 mm), and vancomycin (8.5 mm). The MIC and MBC values of PEO against *S. mutans* were 0.625 and 1.8 μL/mL, respectively. The antibacterial activity of PEO was not influenced by the presence of DMSO in the PEO solution. These results showed that PEO inhibited the proliferation of *S. mutans*.

Figure 1 plots the growth curve of *S. mutans* with or without PEO treatment. The growth of *S. mutans* nearly stops after treatment with the MIC and 2 MIC levels. Furthermore, PEO at 1/2 MIC inhibits the growth rate of *S. mutans* after treatment for 4 h compared to the control group, and its final OD_600_ value was 0.39. This was about half of the control group.

### 3.3. Effect of PEO on Biofilm of S. mutans

#### 3.3.1. Effect of PEO on the Biofilm Biomasses

Since bacterial biofilm plays a crucial role in the development of dental caries, we firstly investigated the effects of PEO at different concentrations on the biofilm biomasses of *S. mutans*. As shown in Figure 2A, the total biofilm biomasses decrease with increased concentrations of PEO, and PEO at 1/4 MIC or above can significantly reduce the biofilm biomasses compared to the control group (*p* < 0.05).

#### 3.3.2. Effects of PEO on Bacterial Activity within Biofilm

To determine the cell activity within the biofilm, we then tested the bacterial activity within the biofilm using an XTT assay. As shown in Figure 2B, the bacterial activity within biofilm significantly decreases with the increase of PEO concentrations, and PEO at 1/4 MIC or above evidently inhibits bacterial activity (*p* < 0.05), which is consistent with the results of the total of biofilm biomasses.

#### 3.3.3. Effects of PEO on the Bacterial Viability within Biofilm

We used the Live/Dead^®^BacLight^TM^ Bacterial Viability kit to observe bacterial viability within the biofilm. SYTO 9 stains all bacterial cells with green fluorescence, and PI stains the cells’ impaired membrane with red fluorescence. In Figure 2C, the red fluorescence increases with increased PEO concentrations, especially at MIC or above, indicating more impaired bacterial cell membranes. From the merged images of live and dead bacteria, we also observed that the color yellow is more significant with the increased PEO concentrations.

#### 3.3.4. Effect of PEO on the Biofilm Structure

SEM can clearly observe the damage degree of bacterial biofilm structure. Figure 2D shows representative images of biofilms after treatment with or without PEO. We observed that *S. mutans* produces large clusters or colonies of bacteria. By contrast, PEO from 1 to 4 MIC treatment changes the bacterial biofilm structure to be sparse and loose; the number of bacteria observed in the region also decreases significantly, and no bacterial colonies are formed.

### 3.4. Effects of PEO on Cell Damage of S. mutans

#### 3.4.1. Effects of PEO on LDH Activity

LDH is an intracellular enzyme of *S. mutans* that catalyzes pyruvate to synthesize lactic acid; LDH activity in the supernatant can be detected when the bacterial cell membranes are incomplete or damaged. As shown in Figure 3A, LDH activities in the supernatant increases with the increase of PEO from 1/8 MIC to 1/2 MIC (*p* < 0.05); LDH activity is positively correlated with PEO concentration, suggesting that PEO can penetrate through the bacterial biofilm and destroy cell membranes, resulting in the release of LDH.

#### 3.4.2. Effects of PEO on Ca^2+^ Leakage

Intracellular Ca^2+^ will leak into the supernatant of the culture when bacterial cell membrane is damaged. As shown in Figure 3B, the Ca^2+^ contents in the supernatant increase with the increase of PEO concentrations, which is consistent with the results of LDH.

### 3.5. Effects of PEO on Cell Adhesion

To discover the underlying reasons for PEO inhibiting bacterial biofilm, we evaluated the effects of PEO on bacterial adhesion. Firstly, the biofilm of *S. mutans* was formed after being cultured for 24 h, and then incubated for another 24 h with PEO treatment. As shown in Figure 4A, the adherence inhibition rate increases after PEO treatment from MIC or above, indicating that PEO significantly reduces bacterial adhesion and dissociates the adhered bacteria.

### 3.6. Effects of PEO on EPSs Content

EPSs are a virulence factor necessary for *S. mutans* to adhere to surface of teeth and form a cariogenic biofilm secreted by GTFs. PEO reduces the contents of EPSs produced by *S. mutans* as measured by the phenol sulfuric acid method. Figure 4B indicates that PEO significantly inhibits water-soluble polysaccharides, compared with the control group (*p* < 0.01,). The inhibition rate of PEO on water-soluble polysaccharides increases significantly with the concentrations from 1/16 MIC to 1/2 MIC. PEO also significantly inhibits water-insoluble polysaccharides, and the inhibitory effects increase with the increase in PEO concentration (*p* < 0.01) (Figure 4C).

### 3.7. Effects of PEO on GTFs Activity

GTFs are the key enzyme to form biofilm. It can synthesize insoluble polysaccharides with sucrose and attach bacteria to the surface of teeth. As shown in Figure 4D, PEO has inhibitory activity against GTFs, resulting in a significant reduction in the synthesis of water-insoluble polysaccharides (*p* < 0.01). The inhibitory effect of PEO is positively correlated with its concentrations. Our results indicate that PEO can prevent bacteria from adhering or aggregating to form biofilm by inhibiting the synthesis of water-insoluble polysaccharides and the activity of GTFs.

### 3.8. Toxicity Analysis of PEO on HOECs

Finally, we also evaluated the cytotoxicity of PEO in human oral epithelial cells (HOECs). The IC_10_ value usually represents a non-cytotoxic concentration. The IC_10_ value of PEO on HOECs was 1.299 μL/mL, which was significantly higher than double of the MIC, indicating that PEO had no cytotoxicity on human oral epithelial cells but significantly inhibited the formation of biofilm of oral pathogens.

## 4. Discussion

The chemical components of propolis are very complex, and more than 600 constituents have been identified from different types of propolis [30]. The precise composition of propolis varies depending on the plants sources, bee species, and geographical locations [31]. Poplar propolis is one of the most important propolis with comprehensively biological activities [13,15,16]. The antibacterial properties of propolis have been widely documented in the scientific literature [12]. Many studies have indicated that propolis has more powerful antimicrobial activities against Gram-positive than Gram-negative bacteria [32]. We also demonstrated that propolis has excellent antibacterial activity against methicillin-resistant *Staphylococcus aureus* [23]. It is generally believed that the antibacterial activity of propolis mainly comprises flavonoids and phenolic acids [33]. However, Bridi et al. (2015) indicated that the concentration of those components does not always correlate with the antibacterial activity observed in vitro [34]. We found that PEO had excellent inhibitory effects on *S. mutans*, and that the main chemical components were himachalen, curcumene, bergamotene, sesquicineole, etc, indicating that propolis has more component-exerting antibacterial activities.

Dental plaque is a kind of oral bacterial biofilm, which is the key virulence factor causing dental caries [35]. Since biofilms increase the drug resistance of bacteria, inhibiting or destroying biofilms is crucial to preventing and treating dental caries. *S. mutans* is one of the main microorganisms attached to the surface of teeth. Many studies have proven that *S. mutans* biofilm is widely used as a dental caries model to study the mechanism of biofilm or drug screening. In the present study we found that PEO suppressed the proliferation of *S. mutans* within biofilms, inhibited the total of biofilms biomasses, and disrupted biofilm structure. The destruction of the bacterial biofilm structure enabled PEO to promote membrane permeability, releasing LDH and calcium ions and inhibiting bacterial proliferation.

The formation of biofilm is a dynamic process that includes three continuous steps: attachment, aggregation and maturation. Attachment is a prerequisite for the formation of biofilm, which is conducive to the development and maturity of biofilm [36]. We also found PEO exhibited excellent inhibitory activities against bacterial adhesion by reducing the production of extracellular polysaccharides, including water-soluble and water-insoluble polysaccharides. Furthermore, one of the most important virulence factors in forming a biofilm is GTFs, which synthesize extracellular polysaccharides with sucrose and attach bacteria to the teeth’s surfaces and was also inhibited by PEO. In short, PEO can suppress the activity of GTFs to reduce the production of extracellular polysaccharides and alleviate bacterial adherence.

Being a natural product used in the oral cavity, the evaluation of its toxicity is particularly important for human health. PEO had no cytotoxicity to human oral epithelial cells. Based on its excellent inhibitory activity against *S. mutans* and no cytotoxicity to normal oral cells, PEO has great potential in preventing and treating oral bacterial infection caused by *S. mutans,* especially dental carie.

## 5. Conclusions

Altogether, our study results showed that PEO has excellent antibacterial activity against *S. mutan*, which works by inhibiting cell viability within the biofilm, by decreasing the total of biofilm biomasses, and by destroying the biofilm structure. We also noted that PEO alleviates bacterial adherence, reducing the production of extracellular polysaccharides by inhibiting the activity of GTFs. Moreover, PEO had no cytotoxicity in normal oral cells. As a result, PEO demonstrates great potential for use in the prevention and treatment of dental caries.

## Figures and Tables

**Figure 1 nutrients-14-03290-f001:**
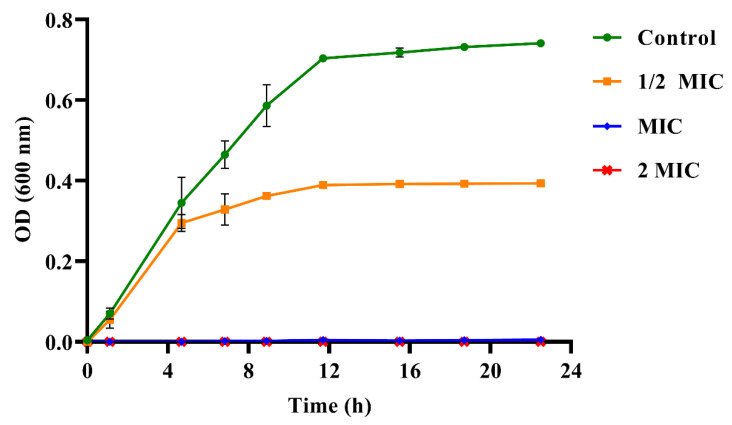
The growth curve of *S. mutans* with or without propolis essential oil (PEO) treatment. Error bars indicate the standard error of the mean; where error bars are not visible, they are smaller than the symbol. 1/2MIC, 0.3125 μL/mL; MIC, 0.625 μL/mL; 2MIC, 1.25 μL/mL.

**Figure 2 nutrients-14-03290-f002:**
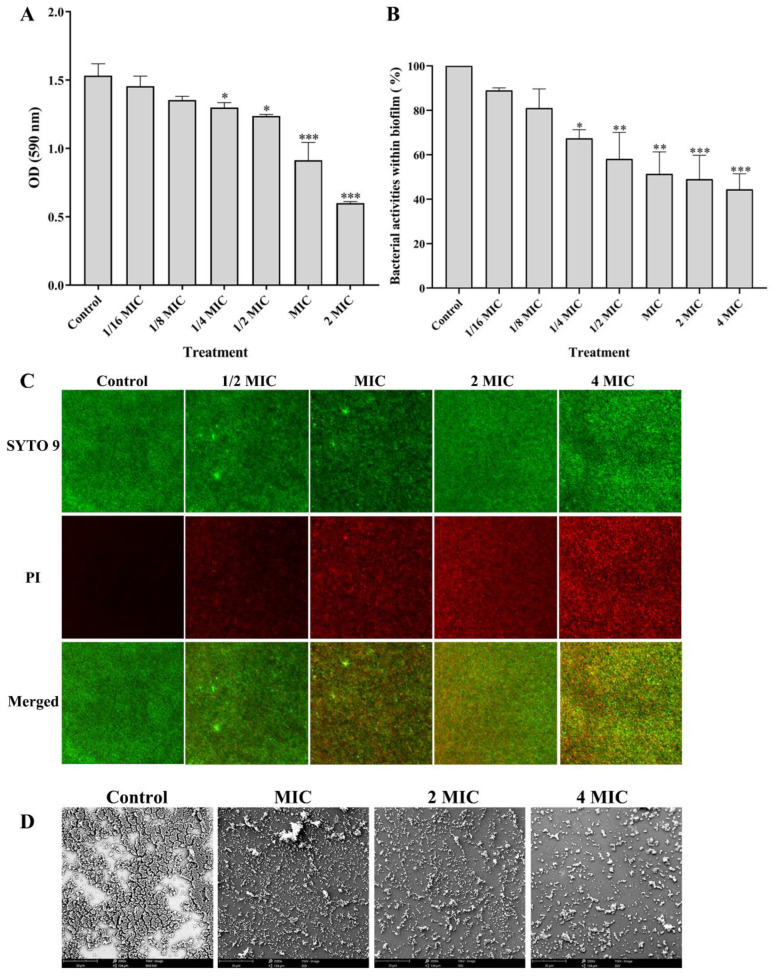
Effect of PEO on biofilm of *S. mutans*. (**A**) PEO suppresses the total biofilm biomasses with the increase of concentration (* *p* < 0.05, ** *p* < 0.01, *** *p* < 0.001). (**B**) PEO decreases the bacterial activities within biofilm with the increase of concentrations. (**C**) Representative fluorescent images of cell viability within biofilm using Live/Dead^®^BacLight^TM^ Bacterial Viability kit. (**D**) SEM-representative images of biofilms after treatment with or without PEO using SEM. 1/4MIC, 0.15625 μL/mL; 1/2MIC, 0.3125 μL/mL; MIC, 0.625 μL/mL; 2MIC, 1.25 μL/mL; 4MIC, 2.5 μL/mL.

**Figure 3 nutrients-14-03290-f003:**
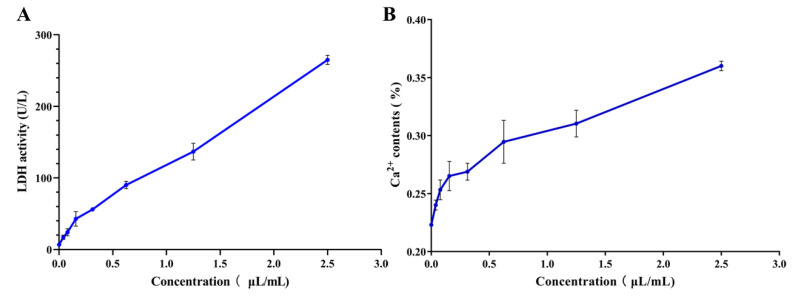
Effects of PEO on cell damage of *S. mutan*. (**A**) PEO increases LDH activities in the supernatant with the increase of concentration. (**B**) PEO increases the Ca^2+^ contents in the supernatant with the increase of concentrations.

**Figure 4 nutrients-14-03290-f004:**
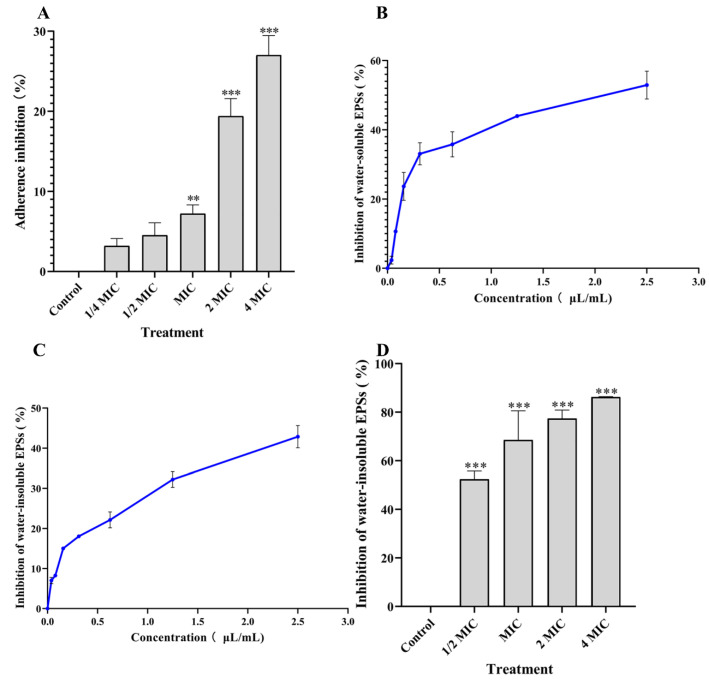
Effects of PEO on cell adhesion, polysaccharides (EPSs) and glucosyltransferases (GTFs). (**A**) PEO decreases bacterial adherence in a concentration-dependent way. (**B**) PEO inhibits water-soluble polysaccharides. (**C**) PEO inhibits water-insoluble polysaccharides. (**D**) PEO depresses the activity of GTFs, and the changes in the water-insoluble polysaccharide content produced were used to indicate PEO’s effect on GTFs (** *p* < 0.01, *** *p* <0.001). 1/4MIC, 0.15625 μL/mL; 1/2MIC, 0.3125 μL/mL; MIC, 0.625 μL/mL; 2MIC, 1.25 μL/mL; 4MIC, 2.5 μL/mL.

**Table 1 nutrients-14-03290-t001:** Chemical constituents of PEO by GC-MS analysis.

No	Compounds	RI ^1^	RT (min) ^2^	PA (%) ^3^
1	Ethyl benzenecarboxylate	1170	16.833	2.4
2	α-Cedrene	1408	26.33	1.14
3	α-Bergamotene	1426	27.305	4.5
4	(E)-β-Famesene	1438	28.15	1.14
5	β-Himachalene	1490	29.108	13.94
6	α-Curcumene	1493	29.247	11.28
7	β-Bisabolene	1505	30.187	3
8	Sesquicineole	1517	30.396	4.35
9	Cadina-1(10)	1527	30.712	2.28
10	α-Copaen-11-ol	1539	31.226	1.67
11	Guaiol	1595	33.547	2.06
12	γ-Eudesmol	1630	34.8	2.2
13	β-Eudesmol	1645	35.468	2.91
14	α-Eudesmol	1652	35.58	1.99
15	Bulnesol	1666	36.015	1.03
16	Bisabolol	1680	36.615	1.01

^1^ RI, retention index relative to n-alkane C_7_-C_30_ on the HP-5MS column. ^2^ RT, retention time. ^3^ PA, percentage of each component peak area in total component peak area.

**Table 2 nutrients-14-03290-t002:** The DIZ, MIC and MBC of PEO against *S. mutans*.

Strain	DIZ (mm) ^1^					MIC (μL/mL)	MBC (μL/mL)
	Control	PEO	Gentamycin Sufate (10 μg/mL)	Ampicillin, Sodium Salt (10 μg/mL)	Vancomycin Hydrochloride (10 μg/mL)		
*S. mutans*	6	24.5 ± 0.71	22.5 ± 2.12	11.0 ± 1.41	8.5 ± 0.71	0.625	1.8

^1 ^DIZ: the value indicated as an average of six replicates ± standard error. DIZ, diameter of inhibition zone. MIC, minimum inhibition concentration. MBC, minimum bactericide concentration.

## Data Availability

Not applicable.

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
