# Peer review of "Anti-Biofilm Activities of Chinese Poplar Propolis Essential Oil against Streptococcus mutans"

_nutrients, 2022, doi:10.3390/nu14163290_

Round 1

Reviewer 1 Report

The article is written as per Journal style.

1. However there are few missing references. These must be updated.

2. The identification of compounds with Kovat index or RRI, the current identification is incorrect. The sequence of elution of compounds are not correct in table 1, which requires major correction (refer literature which describes the analysis of essential oil using HP5MS or DB5MS or equivalent column). Calibrate the instrument with C-7 to C30 alkanes and determine the RRI for identification of constituents.

3. Plagiarism is 34%, It can be reduced to less than 20%. Single source plagiarism can be reduced. 

4. Reference like https://www.mdpi.com/1420-3049/26/16/5076, can be included and consulted for identification of constituents and correct sequence 

5. Refer for plagiarism pdf file ( ak nutrient) and reviewed pdf for comments.

Author Response

  1. However there are few missing references. These must be updated.

Reply: We appreciate the reviewer’s comments. Following the reviewer’s comments, we have added relevant references, which can be found in the manuscript, and the revisions have been highlighted.

  1. The identification of compounds with Kovat index or RRI, the current identification is incorrect. The sequence of elution of compounds are not correct in table 1, which requires major correction (refer literature which describes the analysis of essential oil using HP5MS or DB5MS or equivalent column). Calibrate the instrument with C-7 to C30 alkanes and determine the RRI for identification of constituents.

Reply: We appreciate the reviewer’s comments. Following the reviewer’s comments, we re-analyzed the constituents of PEO, and the results could be found in Table 1 of page 6, line 244 .

  1. Plagiarism is 34%, we would like to ask you to reduce similarity and try to keep it below 30% according to the attachment(according to the suggestion of editor).

Reply: We appreciate the reviewer’s comments. Following the reviewer’s comments, we have revised relevant sentences to reduce plagiarism, and the revisions have been highlighted.

  1. Reference like https://www.mdpi.com/1420-3049/26/16/5076, can be included and consulted for identification of constituents and correct sequence 

Reply: We appreciate the reviewer’s comments. Following the reviewer’s comments, we have consulted the article in reference 21.

  1. Refer for plagiarism pdf file ( ak nutrient) and reviewed pdf for comments.

Reply: We appreciate the reviewer’s comments. We have revised the manuscript according to the suggestion from the reviewer and editor.

Reviewer 2 Report

The present manuscript describes the anti-biofilm activities of Chinese poplar propolis essential oil against Streptococcus mutans, a common cariogenic bacterium

I found the work interesting but some minor revision should be done in order to make the paper suitable for publication.

Introduction

Lines 32-33 “streptococcus mutans (S. mutans), which is considered one of the main pathogens and plays an.”

I suggest replacing by “Streptococcus mutans (S. mutans), being one of the main pathogens that plays an”

Lines 44-46 

I suggest to make two sentences instead of one

Lines 51 

Replace “compounds” by “product”

Lines 54 

Replace “Its antibacterial resistance” by “Its antibacterial activity”

Line 56

Delete “the” after PEO´s 

Materials & Methods

Line 66

Delete “were obtained from”

Line 71

What do you mean by “According to relevant report, the”

Line 86

Replace “The determination method of DIZ was according to the published” by “The method of DIZ was performed according to a published

Lines 88-89

How did you treated the sterile filter paper disks (6 mm) PEO? Didn´t you pipetted the PEO onto the blank discs previously placed and pressed onto the agar plates?

Line 103

Replace “MBC is defined as the minimum concentration of bacterial colonies that cannot be seen” by “MBC is defined as the minimum concentration for which no bacterial colonies were seen”

Line 126

I suggest replacing “detected at” by “read at”

Line 208

Replace “The extraction of GTFs and determination” by “The extraction and determination of GTFs”

Line 221

Add PEO to the tittle: Determination of PEO cytotoxicity to HOEC cells

Line 226

Replace “tested” by “read”

Results

Line 243

Replace “the antibacterial activities of PEO against S. mutans, and the results from DIZ, MIC, and MBC indicated” by “the antibacterial activitiy of PEO against S. mutans and results indicated” 

Line 244

Replace “The DIZ value of PEO was 24.5 mm higher than” by A” The DIZ value of PEO (24.5 mm) was higher than”” 

Line 247

Delete “detected”

Line 290

Caption of Figure 2 

Replace concentrations” by concentration “. Replace “Represent” by “representative”. Change last o: (D) SEM representative images of biofilms after treatment with or without PEO.

Line 309

Caption of Figure 3 

Replace concentrations” by concentration “.

Line 322

Replace “S. mutans using the phenol sulfuric acid method by “S. mutans as measured by the phenol sulfuric acid method”

Line 337

Caption of Figure 4 

Replace “PEO decreases bacterail adherence with the increase of concentrations.” by “PEO decreases bacterial adherence in a concentration-dependent way. Replace “was” by “were” (line 340)

Discussion

Line 352

What do you mean by “comprehensively biological activities?” Anyway, references should be added to this sentence (3-4 reviews perhaps?)

Line 356

Delete “we also”

demonstrated that propolis has excellent antibacterial activity against methicillin resistant Staphylococcus aureus” Others have already shown propolis activity against this bacterium. Please provide the references of such studies

Line 365

Replace “increases” by “increase”

Line 384-385

I suggest:

Being a natural product used in the oral cavity, the evaluation of its toxicity is particularly important for human health.

Author Response

Lines 88-89

How did you treated the sterile filter paper disks (6 mm) PEO? Didn´t you pipetted the PEO onto the blank discs previously placed and pressed onto the agar plates?

Reply: We appreciate the reviewer’s comments. The detailed method that we treated the sterile filer paper disks was as folllows: add 10 μL of PEO onto the sterile filter paper disks (6 mm) until the filter paper has completely absorbed the essential oil, and then place it on the agar plates.

All the other questions raised by the reviewer have been revised, which could be found in the manuscript, and all the revisions made to the manuscript have been highlighted.  

Reviewer 3 Report

Dear Editor and authors

Overall, the paper is well-written, and provides an interesting approach to controlling caries-forming biofilms.

It is well written, structured, and demonstrates in vitro the possibility of using Propolis for a possible caries control.

However I emphasize

Introduction

(1) streptococcus mutans (S. mutans), line 32, change to Streptococcus

Matetials ans methods

(2) lacks references in topics below

    2.3. Gas chromatography-mass spectrometry (GC–MS) analysis of PEO

    2.5.2. Determination of the cell activity within biofilm

    2.6. Cell Damage Assay

    2.7. Determination of bacterial adhesion

    2.8. Extracellular Polysaccharides (EPSs) Production Assay

    2.10. Determination of cytotoxicity to HOEC cells

(3) Also in the item 2.10. (Determination of cytotoxicity to HOEC cells), who is the CCK8 kit manufacter?

Results

(4) There is no information of which concentration in the range of 0.156 to 13 2.5 μL/mL is the MIC

(5) Reformat Table 2, put the parentesis in the same line

(6) Please, change 1/2MIC, MIC, 2MIC in figures 1, 2, 4 for concentration

BEst regards

Author Response

Introduction

  1. streptococcus mutans (S. mutans), line 32, change to Streptococcus

Reply: We appreciate the reviewer’s comments. Following the reviewer’s comments, we have changed this word in line 32.

 Materials and methods

  1. lacks references in topics below

    2.3. Gas chromatography-mass spectrometry (GC–MS) analysis of PEO

    2.5.2. Determination of the cell activity within biofilm

    2.6. Cell Damage Assay

    2.7. Determination of bacterial adhesion

    2.8. Extracellular Polysaccharides (EPSs) Production Assay

2.10. Determination of cytotoxicity to HOEC cells

Reply: We appreciate the reviewer’s comments. Following the reviewer’s comments, we have added relevant references in page 2-5, which can be found in the manuscript, and the revisions have been highlighted.

  1. Also in the item 2.10. (Determination of cytotoxicity to HOEC cells), who is the CCK8 kit manufacturer?

Reply: We appreciate the reviewer’s comments. Following the reviewer’s comments, we have added the kit manufacture in the item 2.10 of page 5, line 229.

  1. There is no information of which concentration in the range of 0.156 to 2.5 μL/mL is the MIC

Reply: We appreciate the reviewer’s comments. MIC can be found in Table 2 of page 6, line 256, and MIC was 0.625 μL/mL.

  1. Reformat Table 2, put the parentesis in the same line

Reply: We appreciate the reviewer’s comments. We have corrected this mistake in Table 2 of page 6, line 256.

  1. Please, change 1/2MIC, MIC, 2MIC in figures 1, 2, 4 for concentration

Reply: We appreciate the reviewer’s comments. Following the reviewer’s comments, we have added relevant concentrations in each figure captions, which can be found in figures 1, 2, 4 (line 267; line 301-302; line 35-351 ).

Round 2

Reviewer 1 Report

The quality of article and presentation improved significantly. The GC-MS analysis is updated as required.  The manuscript can be accepted after minor spelling corrections.

1. replace "vancomyclin" with vancomycin in table and in text.

Author Response

Thank you for your kineness. We have replace "vancomyclin" with vancomycin in table 2 and in text.